# Cultured Microfungal Communities in Biological Soil Crusts and Bare Soils at the Tabernas Desert, Spain

**Isabella Grishkan [1],\*, Roberto Lázaro [2] and Giora J. Kidron [3]**

[1] Institute of Evolution, University of Haifa, 199 Aba Khoushy Ave, Mount Carmel, Haifa 3498838, Israel
[2] Estación Experimental de Zonas Áridas (CSIC), Carretera Sacramento, W/n, La Cañada de San Urbano, 04120 Almería, Spain; lazaro@eeza.csic.es
[3] Institute of Earth Sciences, the Hebrew University of Jerusalem, Givat Ram Campus, Jerusalem 91904, Israel; kidron@mail.huji.ac.il
\* Correspondence: bella@evo.haifa.ac.il

**Abstract:** We examined the variations in microfungal communities from different surface types (cyanobacterial crusts, lichen-dominated crusts, and noncrusted bare surface) at two different positions—north-oriented slope and sun-exposed plain in the Tabernas Desert, Spain. A total of 77 species from 46 genera was isolated using the soil dilution plate method. The studied mycobiota, similar to the majority of desert mycobiotas, was dominated by melanin-containing species. However, in the Tabernas crusts, unlike the studied crusts of the Negev Desert (Israel) and the Tengger Desert (China), melanized fungi with large multicellular spores were much less abundantly represented, while the thermotolerant group, *Aspergillus* spp., remarkably contributed to the communities' structure. Density of microfungal isolates positively correlated with chlorophyll content indicating possible significant influence of organic matter content on fungal biomass. The variations in crust composition, biomass, and the position of habitats were accompanied by the variations in microfungal community structure, diversity level, and isolate densities, with the communities at the plain sun-exposed position being much less variable than the communities at the north-oriented position. The study shows that microclimatic and edaphic factors play an essential role in the development of crust and noncrust microfungal communities, and their structure can be a sensitive indicator of changing environmental conditions at a microscale.

**Keywords:** biological soil crusts; chlorophyll content; diversity level; microfungal communities; species composition

## 1. Introduction

Biological soil crusts (BSCs, biocrusts) are widely distributed components of arid and semiarid ecosystems, which make a significant contribution to soil surface stabilization, hydrological processes, and nutrient cycling, e.g., References [1–4]. Crust type—cyanobacterial, lichen- or moss-dominated—may vary with local microclimates [5] along a precipitation gradient [6] or in accordance with parent material [7]. At the same time, BSCs may also exhibit high microscale variability as a result of variable wetness duration [8].

Free-living microfungi, together with cyanobacteria, heterotrophic bacteria, green algae, lichens, and mosses play an important role in the composition and functioning of the BSCs. Several studies conducted in the deserts and semiarid grasslands, based both on culture-dependent methods [9,10] and on culture-independent molecular approaches [11–17], revealed a rich diversity of crust-inhabiting fungi mainly belonging to the phylum Ascomycota. These studies have characterized the dominant

and frequent taxa of crust fungal communities and some aspects of their association with different crust types and geographical regions.

In the Negev Desert of Israel, we studied cultivable crust-inhabiting mycobiota along a precipitation gradient [18] and within a dune catena at the Nizzana Research Site (NRS) in the Hallamish dune field of western Negev [19]. At the NRS, high microscale variability of BSC types was found, with five crust types defined: Four cyanobacterial crusts and one moss-dominated crust [20]. Significant differences characterized their chlorophyll, protein, and carbohydrate content. The crusts also varied in their daylight surface wetness duration associated with the time in which photosynthesis took place [21], with longer wetness duration resulting in the formation of high-biomass cyanobacterial crusts and in higher cover of lichens and mosses [20].

In both aforementioned mycological studies, the BSC microfungal communities, similar to those of other regions in the Negev, were dominated by melanin-containing species with large multicellular conidia. Abundance of this xeric group increased southward along the precipitation gradient and towards the more xeric crusts within the dune catena at NRS. Density of microfungal isolates was positively correlated with the chlorophyll content, indicating the possible significant influence of organic matter content and wetness duration on fungal biomass [19]. There was similarity between the variations in crust microfungal communities within the dune catena at NRS [19] and those along the precipitation gradient [18] in the Negev Desert, implying that regional climatic variability may have an effect on microfungi comparable to the effect of microclimate.

We also examined the variations in microfungal communities inhabiting four different biocrust types—cyanobacterial, mixed (lichens, green algae, cyanobacteria, and mosses in similar proportions), and two moss-dominated crusts, covering the dune field in the vicinity of the Shapotou Research Station (SRS) in the Tengger Desert, China [22]. The topographically induced differences in abiotic conditions caused variations in species composition, isolate densities, and diversity characteristics of the crust microfungal communities at SRS. Similar to the Negev Desert in Israel, the mycobiota of the crusts in the Tengger Desert was dominated by melanized fungi with large multicellular spores, although with different prevailing species. The substantially higher abundance of thermotolerant fungal species in the crusts of the Tengger Desert in comparison with the Negev Desert was associated with principal differences in the precipitation regimes—winter rains at the Negev Desert and summer rains at the Tengger Desert.

Our present study continues mycological characterization of biological soil crusts in different arid regions and is devoted to microfungal communities of BSC in the Tabernas Desert located in the southeastern part of Spain. Contrary to the Negev and the Tengger research sites, where biocrusts on sandy soil are distributed, in the Tabernas Desert biocrusts occupy fine-grained (silty) soils [23]. Additionally, unlike the Negev and Tengger deserts, BSCs in the Tabernas Desert are dominated by different lichen species [24,25]. The current research was designed to study the variation in the microfungal communities within different crust types (cyanobacterial and lichen-dominated) covering the north-oriented slope and the sun-exposed plain area in comparison with the communities inhabiting adjacent nonbiocrusted bare soils. Based on our previous findings, we hypothesized that the biocrust type in combination with site position would shape the microfungal communities influencing the abundance of their xeric (melanin-containing fungi), thermotolerant (*Aspergillus* spp.), and mesic (*Penicillium* spp.) components. In the course of the study, the following characteristics of the communities were analyzed: Species composition; contribution of major ecological groupings to community structure; dominant groups of species, density of isolates, and diversity level—species richness, the Shannon-Wiener and the evenness indices. The effect of edaphic parameters—chlorophyll content, pH, and electrical conductivity on the above characteristics was also estimated.

## 2. Material and Methods

### 2.1. Site Description

The Tabernas Desert is located in the southeastern part of Spain, Almeria province, in the Rioja-Tabernas basin, surrounded by several Betic ranges: Gador, Nevada, Filabres (all of them higher than 2000 m above sea level (a.s.l.)), and Alhamilla (1387 m). The first three of these ranges intercept most rainfall fronts, which come mainly from the west, thus explaining the low annual precipitation in the Tabernas basin. The study area (240–300 m a.s.l and 20 km distance from the city of Almeria) is formed by a series of parallel catchments on deeply dissected Toronian mudstone of marine origin (Figure 1a). The parent material mainly consists of silt-size (>60%) gypsum-calcareous and siliceous particles, 20–35% of fine sand, and 5–10% of clay [23]. Between some catchments, semiflat areas exist corresponding to old residual hanging pediments. For the dynamic geomorphology of this complex landscape see Alexander et al. [26].

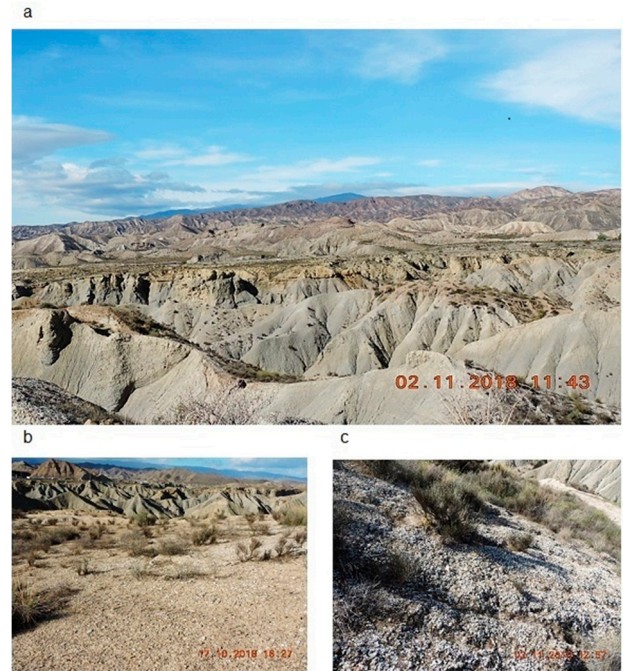

**Figure 1.** (**a**) A general view of the Tabernas basin, (**b**) sun-exposed plain and (**c**) north-oriented slope. Note the dominant white colors of lichens at the north-oriented slope.

The climate of the area is a semiarid-warm Mediterranean with long and hot summers and with an annual average precipitation of 230 mm. There is a high interannual and intra-annual variability of rainfall, mainly concentrated in winter and autumn, with a relatively large number of rainy days per year (40–50). Average annual temperature is 18 °C; average daily maximum during the hottest and coldest months is 34.5 °C and 17.5 °C, respectively, whereas the average of the daily minimums is 4 °C and 19.5 °C in the coldest and the hottest months, respectively [27,28].

In these badlands, vegetation shows a clear pattern: South- to west-facing slopes are normally bare and eroded, while north- to east-facing slopes are covered by grass, dwarf shrubs, and annuals. Eroded landforms occupy a third of the territory, another third is covered by short vascular vegetation with biocrusts in the interspaces, and the rest is covered by biocrusts [24,29].

Four types of biocrust communities were identified in the area [25]. Cyanobacterial crusts, which mainly occupy the sun-exposed habitats, are dominated by *Microcoleus* sp. [30], accompanied by small lichens such as *Collema* spp., *Endocarpon pusillum*, and *Fulgensia* spp. Out of three types of lichen-dominated crusts, the two most frequent mesic crusts are dominated by whitish-colored

*Squamarina lentigera* and *Diploschistes diacapsis* (sometimes accompanied by *Buellia zoharyi*). The third type, mainly occupying the sun-exposed plains and gentle north-facing slopes is characterized by dark-colored *Psora decipiens*, which forms the most xeric lichen-dominated crusts with a mosaic distribution of cyanobacteria and some scarce mosses (*Didymodon* spp. and *Tortula* spp.).

Biocrusts impact soil surface hydrology in the area. They affect both runoff yield [31] and evaporation. The surface of cyanobacterial crusts is characterized by higher daily temperatures in comparison with the surface of lichen-dominated crusts (Figure 2a), accompanied by lower subsurface moisture content (Figure 2b). Cyanobacterial crusts in the area are progressively replaced by lichen crusts (dominated by *S. lentigera* and *D. diacapsis*), with the *Lepraria isidiata* crusts being the most late-successional stage, mainly confined to the stable north-facing slopes [5].

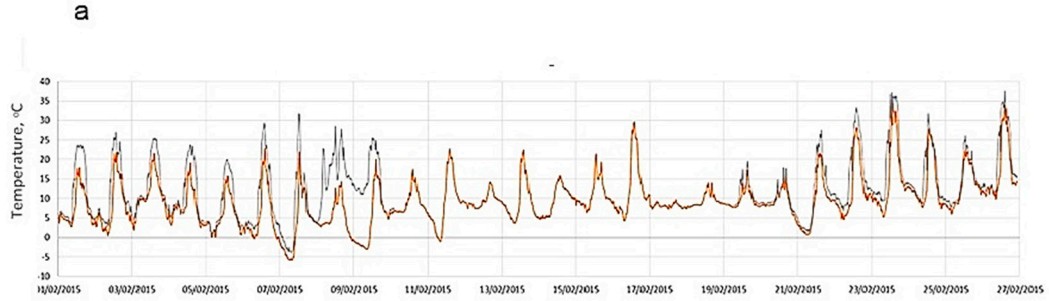

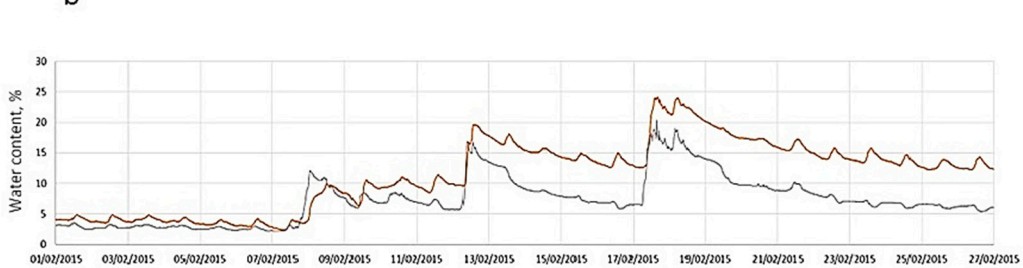

**Figure 2.** (**a**) Surface temperature and (**b**) water content in subcrust soil layer measured during February 2015 at cyanobacterial crusts (black line) and crusts dominated by *Diploschistis diacapsis* (carrot line) at a north-oriented position at the Tabernas Desert.

### 2.2. Sampling

The sampling was designed in order to examine the effect of two environmental factors—habitat position (sun-exposed and north-oriented) and surface type (cyanobacterial crust, crusts dominated by different lichen species, and bare soil) on microfungal communities.

The sampling was carried out during the spring of 2015 at two different positions—the upper-hanged sun-exposed plain and the shaded north-oriented hillslopes (Figure 1b,c). At each position, four crust types were sampled: Cyanobacteria-dominated crusts and three types of lichen crusts—dominated by *S. lentigera, D. diacapsis*, and *P. decipiens*; additionally, samples from the adjacent bare soil were collected. Extremely limited lichen growth at the sun-exposed south-facing slope (due to extensive erosion) did not allow a complete sampling from this slope, and therefore sampling was conducted at the sun-exposed plain. At each position from each surface type (biocrusts and bare soil), four samples were collected at a depth of 0–1 cm (the uppermost crust layer that hosts all photoautotrophic components), from a plot of $7 \times 7$ cm, at a distance of 1–1.5 m from each other, 40 samples altogether. The samples, placed in sterile paper bags, were stored in dry conditions until processing.

### 2.3. Characterization of Fungal Communities

For isolation of microfungi, the soil dilution plate method [32] was employed. Despite certain limitations and biases [33], it remains a useful approach for the characterization of the ecology of fungal communities [34]. The method is especially applicable to desert soils where microfungi may exist for a long period in a dormant (spore) state.

Ten grams of soil samples were used in a dilution series. Two culture media with different C and N sources were employed: Malt Extract Agar (MEA) and Czapek's Agar (CzA) (Sigma-Aldrich Inc, St. Louis, MO, USA). Streptomycin (Spectrum Chemical Mfg. Corp, Gardens, USA) was added to each medium (100 μg/mL) to suppress bacterial growth. Soil suspension in an amount of 0.5–1 mL from the dilutions 1:100 (soil:sterile water) was mixed with the agar medium at 40 °C in Petri dishes (90-mm diameter). The plates were incubated at 25 °C in darkness for 10–15 days (three plates for each medium).

After incubation, all emerging colonies were examined under binocular (magnification of 45×) and divided into morphotypes according to the colony appearance (texture, color, size, etc.) and, whenever it was possible, presence of a definite kind of sporulation (fruit bodies or spores). Each morphotype was transferred to MEA and CzA for purification and further taxonomic identification. In an attempt to induce sporulation, all nonsporulating isolates were also grown on Oatmeal Agar (Sigma-Aldrich Inc, St. Louis, MO, USA) as recommended by Bills et al. [34], and on Water Agar (20 g agar, 1000 mL water). Taxonomic identification was based mainly on the microscopic examination of morphological characteristics of sporulation—fruit bodies, sporophores, and spores (their shape, size, color, and mode of aggregation). Numerous identification books were used, for example, "Compendium of Soil Fungi" [35], "Dematiaceous Hyphomycetes" [36], and "Identification of Common *Aspergillus* Species" [37]. Two consistently isolated nonsporulating strains were subjected to molecular identification performed at Hy Laboratories Ltd., Rehovot, Israel. DNA was extracted from pure cultures according to the procedure described by Graham et al. [38], and each of the 18S-28S rRNA regions were amplified by polymerase chain reaction (PCR) using Hy-FID PCR kit (Cat No. 505, Hylabs, Israel). The resulting amplicons were sequenced using the ABI BigDye V1.1 Terminator Cycle Sequencing kit (Applied Biosystems) and an ABI 3730 automated DNA gene analyzer according to the manufacturer's instructions. Obtained sequences were aligned and compared to sequences available at the National Centre for Biotechnology Information (NCBI) (http://www.ncbi.nlm.nih.gov) using BLASTN search [39]. All names of the identified species are cited according to the Species Fungorum database (www.speciesfungorum.org).

### 2.4. Measurement of Chlorophyll Content, pH, and Electrical Conductivity (EC)

One centimeter-depth cores, 1 cm$^2$ each, were sampled from the crusts and bare soil. For each surface type, six replicates were used for the measurement. Concentration of chlorophyll *a* (hereafter chlorophyll) was determined using extraction by hot methanol (70 °C, 20 min) in the presence of MgCO$_3$ (0.1% *w/v*) in sealed test tubes. Chlorophyll concentration was calculated in accordance with Wetzel and Westlake [40]. The pH and EC measurements were performed in a water paste using Cyberscan pH11 and Cyberscan con11, respectively (EuTech Instruments). To create the paste, distilled water was added to 20 g of soil and left for 2 h to facilitate equilibrium between the dissolved salts and the water.

### 2.5. Data Analyses

Density of fungal isolates was expressed as "colony forming units" (CFU) per gram dry substrate. Relative abundance of species was calculated as number of isolates of a particular species in the sample/total number of all isolates in the sample. Analysis of diversity was based on the Shannon-Wiener index (*H*) and evenness ($J = H/H_{max}$) [41].

To analyze spatial variations in the microfungal community structure, three major groupings were chosen: *Penicillium* spp. consisting mainly of mesophilic fungi, *Aspergillus* spp., containing mainly thermotolerant and thermophilic fungi, and the major desert group of melanin-containing fungi; in the latter grouping, melanized species with large multicellular spores were examined separately. The contribution of each group to mycobiota structure was estimated as an average of its relative abundance in the sample and its percentage in the Shannon index. We used the latter characteristic together with direct relative abundance because (i) it is logarithmic, thus preventing an overestimation of heavily sporulating species, and (ii) it takes into account not only the number of isolates but also the number of species comprising the aforementioned groupings.

Statistical analysis was conducted using XLSTAT (http://www.xlstat.com) and PAST (http://folk.uio.no/ohammer/past/ for PERMANOVA and the sample rarefaction test). The one-way PERMANOVA analysis based on Euclidean distance was performed to examine the differences in fungal community composition among different surface types (crust types and bare soil) and habitat positions (north-oriented and plain sun-exposed). The absolute abundances of fungi were square root-transformed prior to the analysis. We employed a one-way ANOVA followed by multiple-comparison tests to compare data from different surface types and positions on diversity characteristics, contribution of microfungal groupings, and isolate densities. The relationship of these data with edaphic parameters (chlorophyll content, pH, and electrical conductivity) was estimated by linear regression analyses. A two-way unbalanced ANOVA with interactions was used to test the effect of different environmental factors (surface type and habitat position), separately and in interaction, on the above mycological parameters. To evaluate similarity between communities from different surface types and aspects, the clustering of the communities based on species relative abundances was made by the unweighted pair-group average method with Chi squared distance as the distance coefficient.

## 3. Results

### 3.1. Edaphic Characteristics

Chlorophyll content exhibited a clear pattern, being higher at the north-oriented position than at the plain sun-exposed position. Moreover, it displayed a general increase from the bare surfaces (<10 mg m$^{-2}$), through the cyanobacterial crusts (19–46 mg m$^{-2}$) and the lichen-dominated crusts (98–350 mg m$^{-2}$) (Figure 3a). Expectedly, the chlorophyll content in the most xeric lichen crusts dominated by *P. decipiens* was significantly lower than that in the mesic lichens crusts dominated by *S. lentighera* and *D. diacapsis*.

Except for the crusts dominated by *D. diacapsis*, the soil of all other habitats was alkaline, with pH > 7 (Figure 3b). In the bare surface, pH was higher at the north-oriented positon, while in the crusts, higher pH values were recorded at the plain south-exposed position. All habitats had very low EC, <1.0 dS/m, except for the cyanobacterial crusts at the north-oriented position, with EC being slightly higher than 1.0 dS/m (Figure 3c).

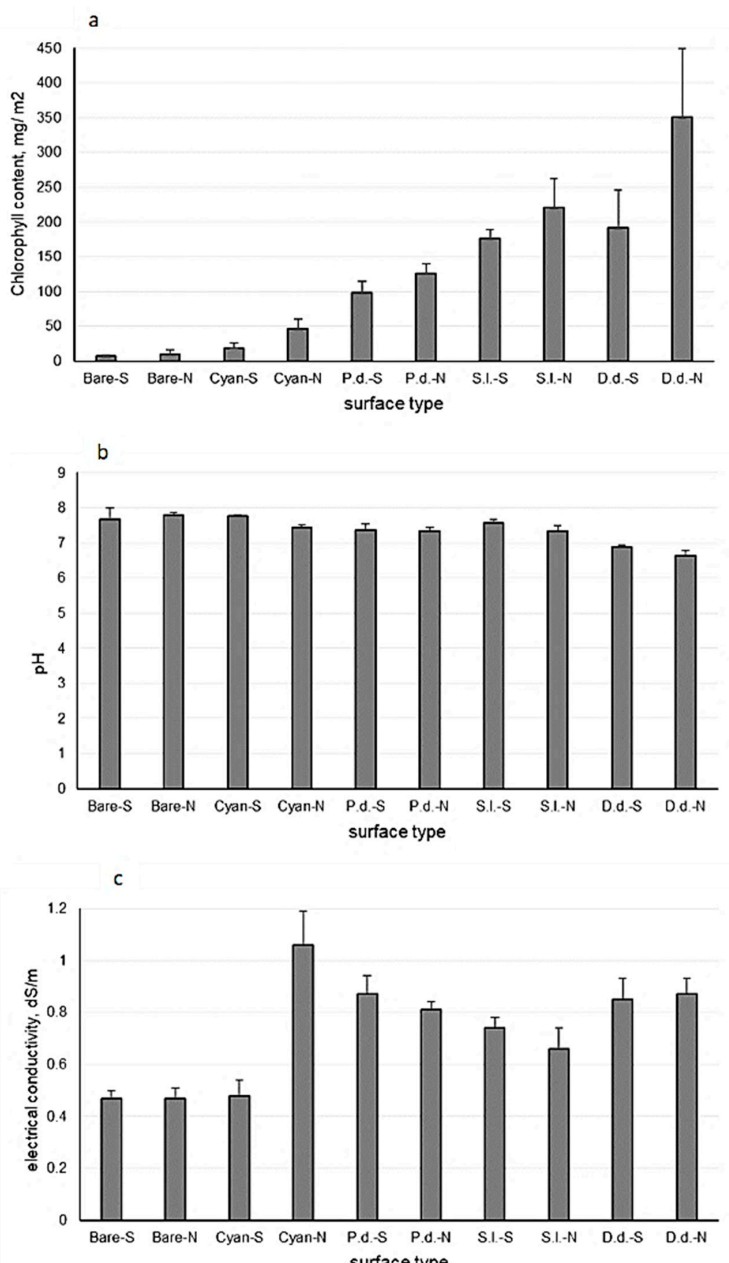

**Figure 3.** (**a**) Chlorophyll *a* content, (**b**) pH, and (**c**) electrical conductivity in different crust types and bare surface at the Tabernas Desert; P.d.—crusts dominated by *Psora decipiens*; S.l.—crusts dominated by *Squamarina lentighera*; D.d.—crusts dominated by *Diploschistis diacapsis*; cyan—cyanobacterial crusts; N—north-oriented position; S—plain sun-exposed position. Vertical bars represent standard deviations (n = 6).

## 3.2. Density of Microfungal Isolates

Isolate densities in the communities at the north-oriented position were 1.1–2.1-fold higher than in the communities at the plain sun-exposed position (Figure 4). Among the biocrust types, crusts dominated by *D. diacapsis* and cyanobacterial crusts at both positions harbored the highest and the lowest CFU numbers, respectively. Microfungal communities from the bare surfaces both at the plain sun-exposed position and at the north-oriented position contained the lowest densities of isolates, which were significantly lower than in the crusted surfaces (Figure 4).

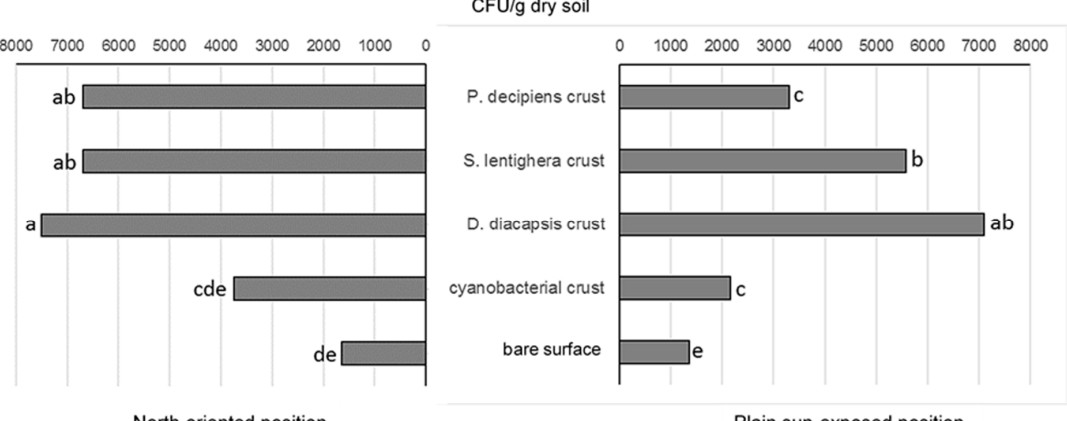

**Figure 4.** Density of microfungal isolates in different crust types and bare surface at the Tabernas Desert. Means with the same letters are not significantly different (a one-way ANOVA, the Fisher test, at the 5% level).

## 3.3. Composition and Diversity of Microfungal Communities

Altogether, 77 species were isolated from Zygomycota (6 species), teleomorphic (morphologically sexual) Ascomycota (11), anamorphic (asexual) Ascomycota (59), and Basidiomycota (1). The species belonged to 46 genera; the most common were *Penicillium* (10 species), *Aspergillus* (11), *Chaetomium*, and *Alternaria* (4 each) (Table 1). Five types of microfungal strains remained nonsporulating in culture and have not been identified. Overall, nearly 1300 microfungal isolates were recovered from the crust and bare soil samples.

**Table 1.** Microfungi in different crust types and bare surface at the Tabernas Desert, with their average relative abundance (%). Underlined species are melanin-containing.

| Species | North-Oriented Positon | | | | | Plain Sun-Exposed Position | | | | |
|---|---|---|---|---|---|---|---|---|---|---|
| | Bare | Cyan [a] | P.d. | S.l. | D.d. | Bare | Cyan | P.d. | S.l. | D.d. |
| Zygomycota | | | | | | | | | | |
| *Absidia corymbifera* | - | - | - | - | 0.3 | - | - | - | - | - |
| *Actinomucor elegans* | - | - | - | - | - | - | - | - | - | 0.1 |
| *Mortierella humilis* | 0.5 | - | - | - | - | - | - | - | - | - |
| *Mucor hiemalis* | - | - | - | - | - | - | - | - | 0.15 | - |
| *M. plumbeus* | - | - | - | - | - | - | - | - | - | 0.1 |
| <u>*Rhizopus arrhizus*</u> | 0.5 | 0.5 | 2 | - | - | 0.9 | 0.6 | 0.75 | 0.45 | 1.4 |
| teleomorphic Ascomycota | | | | | | | | | | |
| *Aspergillus nidulans* | 0.5 | - | - | - | - | - | 0.3 | - | 1 | - |
| *A. rugulosus* | - | - | - | 0.12 | - | - | - | - | - | - |
| <u>*Canariomyces notabilis*</u> | 0.5 | - | - | - | - | - | - | - | - | - |
| <u>*Chaetomidium subfimeti*</u> | - | - | - | 0.12 | 0.1 | - | - | - | - | - |
| <u>*Chaetomium strumarium*</u> | - | - | - | - | 0.4 | - | - | - | - | - |
| <u>*Ch. succineum*</u> | - | 0.35 | 0.2 | - | - | 0.3 | 4.5 | - | - | - |
| <u>*Chaetomium* sp.</u> | 1.5 | - | 0.45 | 0.12 | 0.1 | 0.6 | 1.2 | - | 0.15 | - |
| <u>*Chaetomium* sp.1</u> | - | 0.17 | 0.2 | - | - | - | 0.3 | 0.6 | - | - |
| <u>*Sordaria fimicola*</u> | - | - | - | - | 1.1 | - | - | 0.2 | - | - |
| <u>*Sporomiella minima*</u> | 1.5 | 0.7 | - | - | - | - | 0.6 | - | - | - |
| <u>Immature fruit bodies</u> | - | - | - | - | - | - | 0.3 | 0.2 | - | - |

**Table 1.** *Cont.*

| Species | North-Oriented Positon | | | | | Plain Sun-Exposed Position | | | | |
|---|---|---|---|---|---|---|---|---|---|---|
| | Bare | Cyan [a] | P.d. | S.l. | D.d. | Bare | Cyan | P.d. | S.l. | D.d. |
| anamorphic Ascomycota | | | | | | | | | | |
| *Acremonium charticola* | - | - | 0.2 | - | - | - | - | - | - | - |
| *Alternaria alternata* | 2.1 | 0.5 | 0.35 | 0.25 | 0.1 | 5.2 | 0.3 | 0.6 | 0.6 | 0.9 |
| *A. atra* | 0.5 | 2.2 | 0.2 | 1.2 | 2.0 | 1.5 | 1.2 | 3.4 | 1.5 | 1.4 |
| *A. phragmospora* | 29.5 | 7.4 | 21.9 | 1.3 | 20.9 | 3.1 | 20.2 | 23.2 | 33.2 | 6.5 |
| *A. raphani* | - | - | - | - | - | 0.3 | 0.9 | - | - | 0.1 |
| *Amerosporium concinnum* | - | - | - | - | - | - | - | - | 0.15 | - |
| *Aphanocladium album* | - | 1.3 | 0 | 0.25 | - | - | 0.6 | 0.4 | - | - |
| *Aspergillus alliaceus* | - | - | - | - | - | - | - | - | 0.3 | - |
| *A. flavus* | - | 0.5 | - | 0.12 | - | - | - | 0.2 | - | 0.1 |
| *A. fumigatus* | 42.4 | 2.2 | 48 | 13.4 | 6.4 | 58.7 | 37.5 | 22.8 | 29.5 | 43.6 |
| *A. niger* | - | - | 0.35 | 0.25 | - | 7.4 | - | - | 0.45 | 0.5 |
| *A. niveus* | - | - | - | 0.12 | - | - | - | - | 0.3 | - |
| *A. puniceus* | - | - | - | - | - | - | - | 0.2 | - | - |
| *A. terreus* | - | - | - | - | - | - | - | - | - | 0.1 |
| *A. versicolor* | 0.5 | - | - | - | - | - | - | - | - | - |
| *Aspergillus* sp. | - | - | - | - | - | - | - | - | 0.3 | - |
| *Aureobasidium pullulans* | - | - | - | - | 0.1 | - | - | - | - | - |
| *Beauveria bassiana* | - | - | - | 0.25 | - | - | - | - | - | - |
| *Boeremia exigua* | 7.3 | 5.9 | 14 | 46.6 | 48.3 | 7.1 | 3.9 | 23.2 | 8.2 | 11.4 |
| *Camarosporium aequivocum* | 2.6 | - | - | - | - | 0.3 | 0.3 | 2.6 | - | - |
| *Cladosporium cladosporioides* | - | - | - | - | - | 0.6 | - | - | - | - |
| *Coleophoma empetri* | 0.5 | - | 6.3 | 4.6 | 13.4 | 10.4 | 0.6 | 14.5 | 7 | 16.1 |
| *Curvularia inaequales* | - | 6.4 | 1.6 | - | - | - | - | - | - | 0.9 |
| *Engyodontium album* | - | 0.8 | - | - | - | - | - | - | - | - |
| *Epicoccum nigrum* | 0.5 | - | - | - | - | 0.3 | - | - | - | - |
| *Fusarium gibbosum* | - | 1.3 | - | 4.1 | 1.7 | 0.3 | 2.1 | 1.3 | 0.45 | 1.4 |
| *F. oxysporum* | 4.1 | 0.35 | - | 0.5 | - | 0.9 | 8.4 | - | - | - |
| *F. solani* | | | | | 0.5 | | | | | |
| *Lecanicillium psaliotae* | - | - | - | - | - | - | - | 0.6 | - | - |
| *Malbranchea pulchella* | 0.5 | - | - | - | - | - | - | - | - | - |
| *Metarhizium marquandii* | - | - | - | - | - | - | - | 0.2 | 0.6 | - |
| *Neoscytalidium dimidiatum* | - | 0.17 | - | - | - | - | - | - | - | - |
| *Papulaspora pannosa* | 1.0 | - | - | 0.4 | 0.6 | - | - | - | - | - |
| *Paraboeremia putaminum* | - | 53.9 | - | 0.25 | - | - | 1.2 | - | 1 | - |
| *Penicillium aurantiogriseum* | - | 2.9 | 0.2 | 2 | - | - | 0.9 | - | 0.15 | - |
| *P. brevicompactum* | - | - | - | - | - | - | 0.6 | - | - | 0.1 |
| *P. glabrum* | - | - | - | - | - | - | - | - | - | 0.9 |
| *P. griseoroseum* | - | - | - | - | - | - | - | - | - | 1.7 |
| *P. corylophylum* | - | - | - | - | - | - | - | 0.2 | - | - |
| *P. herquei* | - | - | - | - | - | - | - | - | - | 0.2 |
| *P. janczewskii* | - | - | - | - | - | - | 0.3 | - | - | - |
| *P. lividum* | - | - | - | 0.12 | - | - | - | - | - | - |
| *P. simplicissimum* | - | - | 0.8 | - | - | 0.3 | 0.9 | - | 0.45 | - |

**Table 1.** *Cont.*

| Species | North-Oriented Positon | | | | | Plain Sun-Exposed Position | | | | |
|---|---|---|---|---|---|---|---|---|---|---|
| | Bare | Cyan [a] | P.d. | S.l. | D.d. | Bare | Cyan | P.d. | S.l. | D.d. |
| anamorphic Ascomycota | | | | | | | | | | |
| *P. waksmani* | - | - | - | - | - | - | 0.3 | - | - | - |
| *Phoma medicaginis* | - | - | - | 0.25 | - | - | - | - | - | 0.2 |
| *Phomatodes nebulosa* | - | - | 0.35 | - | - | - | - | - | - | - |
| *Pithomyces africanus* | - | 1.2 | - | - | - | - | - | - | - | - |
| *Plenodomus tracheiphilus* | - | - | - | 1.1 | - | - | - | - | - | - |
| *Pleospora tarda* | 0.5 | 0.35 | - | - | - | - | 0.3 | - | - | 6.9 |
| *Pseudogymnoascus pannorum* | - | 0.7 | - | - | - | - | - | - | - | - |
| *Pyrenochaeta cava* | 1.0 | 0.8 | - | 1.5 | 0.3 | - | 2.7 | 1.1 | 1.5 | - |
| *Schizothecium inaequale* [b] | 1.5 | 6.9 | 1.4 | 0.75 | - | - | 2.7 | 0.9 | 6 | 2.7 |
| *Setophoma terrestris* [b] | 0.5 | 1.0 | - | - | - | 0.9 | 1.5 | - | - | - |
| *Stachybotrys chartarum* | - | 0.17 | - | 1.1 | - | 0.6 | 0.3 | 0.75 | 0.75 | - |
| *Talaromyces purpurogenus* | - | - | - | 0.75 | - | - | - | - | - | 0.1 |
| *T. variabilis* | - | - | - | - | 0.1 | - | - | 0.2 | - | - |
| *Trichoderma koningii* | - | 0.8 | - | - | 0.4 | - | - | - | - | - |
| *T. viride* | - | - | - | 0.6 | - | - | - | - | - | - |
| *Xepicula leucotricha* | - | - | - | 16 | - | - | - | - | 4.6 | - |
| Basidiomycota | | | | | | | | | | |
| *Disporotrichum dimorphosporum* | - | - | - | - | - | - | 2.7 | - | - | - |
| Non-sporulating strains | | | | | | | | | | |
| light-colored | - | 0.5 | - | - | - | - | 1.8 | 0.6 | 0.6 | 0.9 |
| dark-colored | - | - | 1.5 | 1.7 | 3.2 | 0.3 | - | 1.3 | 0.6 | 1.7 |

[a] Abbreviations as in Figure 3; [b] identified by molecular analysis, 98% of maximal identity.

At each position, the species composition of microfungal communities from different surface types was found to be significantly distinct (the one-way PERMANOVA test, $p = 0.0001$). At the north-oriented positon, the pairwise comparisons revealed significant differences between the communities from all surface types ($p < 0.05$) except for the differences between mesic lichen-dominated crusts with *D. diacapsis* and *S. lentighera* ($p = 0.054$). At the plain sun-exposed position, the communities from bare soil, cyanobacterial crusts, and xeric lichen-dominated crusts with *P. decipiens* significantly differed from the mesic lichen crust communities ($p < 0.05$); other differences between surface types were nonsignificant being the least pronounced between bare soil and cyanobacterial crusts ($p = 0.2$). The comparisons between microfungal communities from the same surface type at different position (north-oriented and plain sun-exposed) revealed nonsignificant differences between the bare soil ($p = 0.084$) and the crusts dominated by *S. lentighera* ($p = 0.06$); other differences between the communities were statistically significant ($p < 0.05$).

Cluster analysis based on species relative abundances revealed no clear groups on either surface type or its position. The communities at the plain sun-exposed position formed much more homogenous groups than the communities from the north-oriented position (Figure 5). Two communities at the north-oriented position clustered apart from the others apparently due to the dominance of a pycnidial fungus, *Paraboeremia putaminum* (cyanobacterial crusts), and the prevalence of another pycnidial fungus, *Boeremia exigua*, accompanied by the abundant development of a sporodochial species, *Xepicula leucotricha* (Table 1).

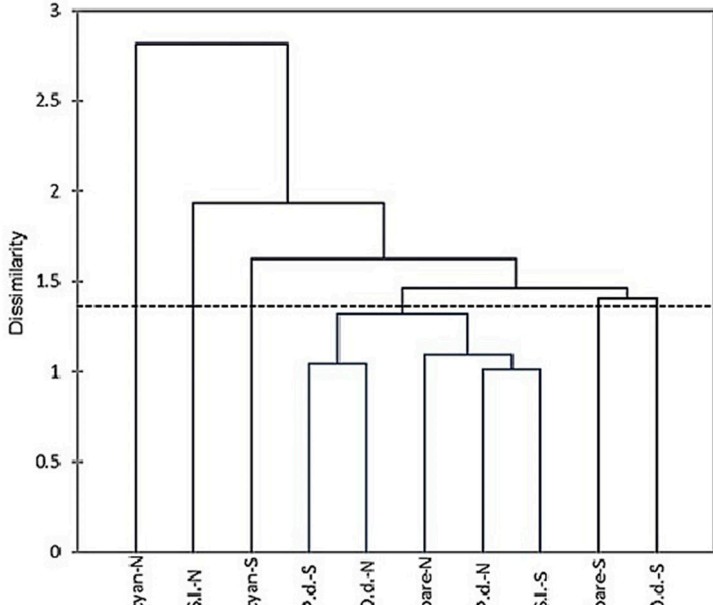

**Figure 5.** Clustering the microfungal communities from different crust types and bare surface at the Tabernas Desert, based on species relative abundance. Abbreviations as in Figure 3.

Nearly the same number of microfungal species, 55 and 56, were isolated from the habitats at the plain sun-exposed positon and the north-oriented position, respectively. Sample-based rarefaction curves reached an asymptote suggesting that the isolation procedure recovered most of fungal species within the communities (Figure 6). All diversity characteristics in the crust communities—species richness, heterogeneity, and evenness, were significantly higher at the plain sun-exposed position as compared to the north-oriented position (except for species richness in the crusts dominated by *S. lentighera*); in the bare surface, an opposite pattern was found (Table 2). At each position, the microfungal communities in the cyanobacterial crusts were more heterogeneous and even than the communities in the lichen-dominated crusts. On the diversity level as a whole, the crust communities at the plain sun-exposed position were more homogenous than the communities at the north-oriented position.

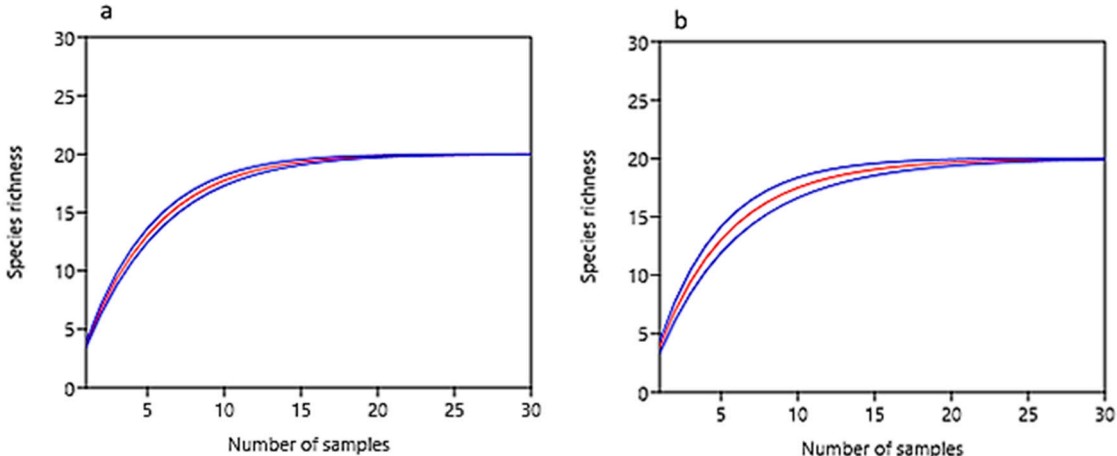

**Figure 6.** Sample-based rarefaction curves for microfungal communities from (**a**) north-oriented position and (**b**) plain sun-exposed position at the Tabernas Desert (at 95% confidence).

**Table 2.** Diversity characteristics of soil microfungal communities (*R*—number of species, including sterile strains, *H*—Shannon index, *J*—evenness) in different crust types and bare surface at the Tabernas Desert. Means with the same letter are not significantly different (a one-way ANOVA, the Fisher test, at the 5% level).

| Surface Type | North-Oriented Position | | | Plain Sun-Exposed Position | | |
|---|---|---|---|---|---|---|
| | *R* | *H* | *J* | *R* | *H* | *J* |
| *P. decipiens* crust | 19 c | 1.56 c | 0.53 ab | 24 ab | 2.02 ab | 0.63 a |
| *S. lentighera* crust | 29 a | 1.86 abc | 0.55 ab | 27 a | 2.0 ab | 0.61 a |
| *D. diacapsis* crust | 19 c | 1.53 c | 0.52 ab | 25 ab | 1.95 ab | 0.61 a |
| cyanobacterial crust | 27 ab | 1.92 ab | 0.58 a | 31 a | 2.27 a | 0.66 a |
| bare surface | 22 b | 1.79 bc | 0.59 a | 20 c | 1.59 c | 0.53 ab |

As Figure 7 shows, melanin-containing microfungi predominated in the majority of the communities; they were represented by 46% of the total species number and by 41.4–96.3% of the contribution index (the species are underlined in Table 1). Species with large (>20 μm) multicellular conidia (mostly *Alternaria phragmospora, A. alternata,* and *A. atra*) comprised less than half of melanized microfungi (1.5–43.8%). In parallel, species with pycnidial fruit bodies (*Boeremia exigua, Coleophoma empetri*, and *Paraboeremia putaminum*) were frequently and abundantly isolated from the majority of habitats (Table 1). Contribution of melanin-containing microfungi was subjected to remarkable spatial variations at the north-oriented position, while at the plain sun-exposed position, the contribution of both melanin-containing species on a whole and melanics with large many-celled spores were much more homogenous (Figure 7).

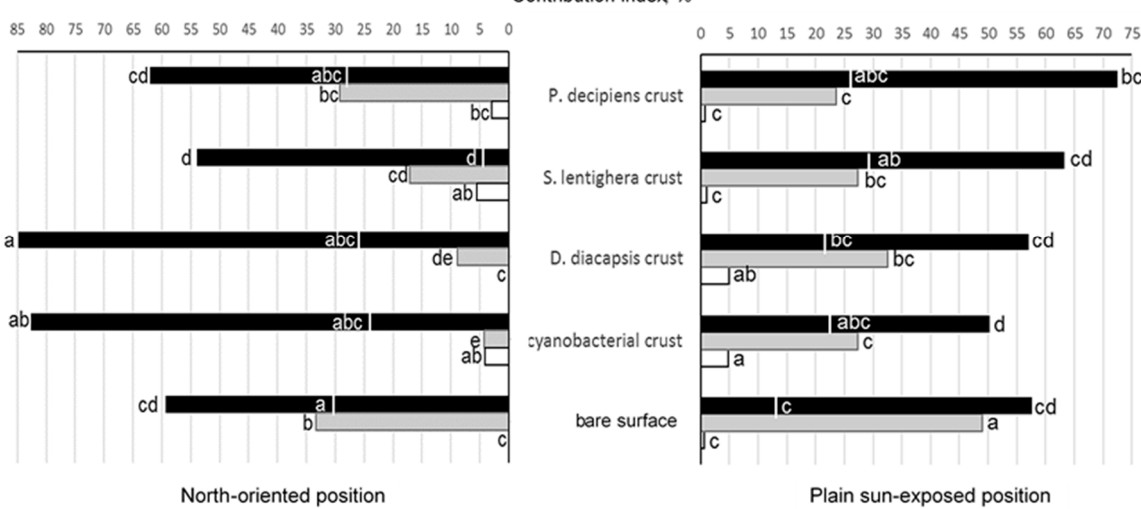

**Figure 7.** Contribution of main microfungal groupings to communities' structure in different crust types and bare surface at the Tabernas Desert: ▉—melanin-containing microfungi; ▢—*Aspergillus* spp.; ▢—*Penicillium* spp.; the area left (plain sun-exposed position) or right (north-oriented position) from the line on the bars of melanin-containing microfungi indicates contributions of species with large multicellular spores. Means with the same letters are not significantly different (a one-way ANOVA, the Fisher test, at the 5% level).

Contribution of the genus *Aspergillus* (with *A. fumigatus* as the most frequent and abundant species) also varied in a lesser degree at the plain sun-exposed position, while at the north-oriented position, its variations were much more pronounced, with minimal (cyanobacterial crusts) and maximal (bare surface) values differing by almost eight-fold (Figure 7). Except for the crusts dominated by *P. decipiens*, contribution of aspergilli was significantly higher in the communities at the plain sun-exposed position

in comparison with the communities at the north-oriented position. At each position, the contribution of aspergilli peaked in the bare surface and was significantly higher than in the crusts. Notably, the contribution of *Aspergillus* spp. was the lowest in the cyanobacterial crusts and crusts dominated by *D. diacapsis* at the north-oriented position where the contribution of melanin-containing species was the highest (Figure 7). The genus *Penicillium* was a minor component of all microfungal communities (0–5.7%), with no consistent pattern of its spatial distribution (Figure 7).

### 3.4. Effect of Surface Type and Orientation on Characteristics of Microfungal Communities

Among diversity characteristics, only species richness was significantly influenced by one of the environmental aspects—surface type and by the interaction between surface type and position (north-oriented or plain sun-exposed) (Table 3). Among microfungal groupings, the contribution of *Aspergillus* spp. was most affected by each aspect separately. Melanin-containing microfungi were also significantly dependent in their distribution both on surface type and position. The contribution of all microfungal groupings, except for aspergilli, was significantly influenced by the cumulative effect of surface type and position (the contribution of melanized species with large multicellular spores was only influenced by the interaction of these aspects). Both environmental aspects separately, but not in interaction, significantly affected the density of microfungal isolates. Between the environmental aspects, the surface type influenced the characteristics of microfungal communities in a stronger way (Table 3).

**Table 3.** Data of two-way unbalanced ANOVA analysis for the effect of surface type (crusts dominated by different lichen species, cyanobacterial crusts, and bare surface), habitat position (north-oriented and plain sun-exposed), and interactions between them on different parameters of microfungal communities at the Tabernas Desert.

| Parameter | Surface Type | Orientation | Type × Orientation |
|---|---|---|---|
| Species richness | 3.17 [@] | NS | 4.05 * |
| Shannon index | NS | NS | NS |
| Evenness | NS | NS | NS |
| Melanin-containing spp. | 2.98 [@] | 4.29 [@] | 3.52 [@] |
| Melanics with multicellular spores | NS | NS | 4.38 * |
| *Penicillium* spp. | 3.95 [@] | NS | 3.44 [@] |
| *Aspergillus* spp. | 7.71 *** | 8.96 ** | NS |
| Isolate density | 16.6 **** | 6.9 * | NS |

[@] ≤0.05; * ≤0.01; ** ≤0.005; *** ≤0.001; **** ≤0.0001.

### 3.5. Relationships of Mycobiotic Characteristics with Edaphic Parameters

Isolate density significantly and positively correlated with chlorophyll content and negatively with pH ($p = 0.001$ and $p = 0.005$, respectively; Figure 8a,b). The contribution of melanin-containing species and aspergilli exhibited significant relationship with EC—positive ($p = 0.01$) and negative ($p = 0.01$), respectively (Figure 8c,d).

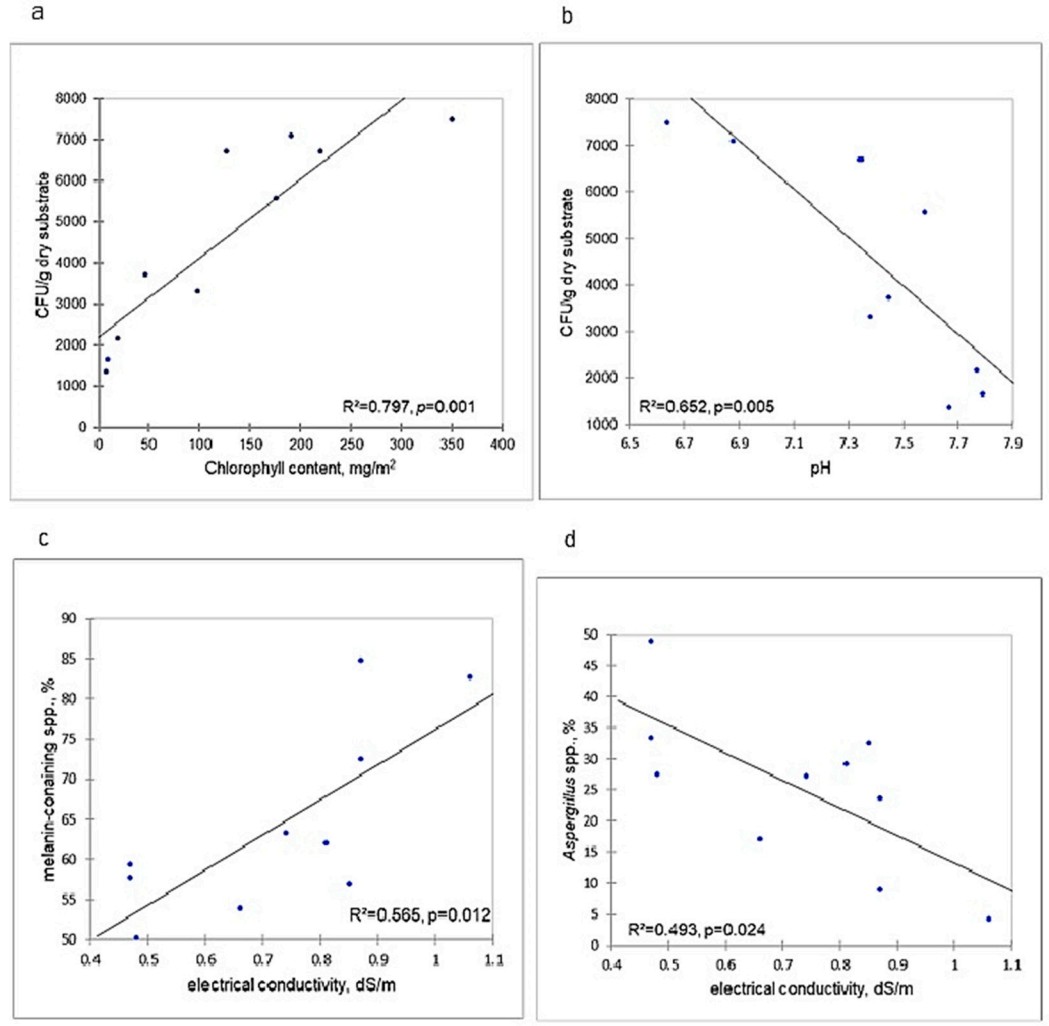

**Figure 8.** Relationship between characteristics of microfungal communities and measured edaphic parameters in different crust types and bare surface at the Tabernas Desert. CFU: Colony forming units.

## 4. Discussion

Expectedly, the general picture of fungal species composition in the biocrusts and noncrusted bare surface from the Tabernas Desert was similar to that of other deserts. This was mainly due to the dominance of melanin-containing species both in number and abundance, which is considered a typical attribute for almost all studied arid soils, e.g., References [42–48]. Such dominance was also found both in the crusts of the Israeli Negev Desert [19] and the Tengger Desert, China [22].

For the crust microfungal communities in Israel and China, the prevalence of melanized species with large multicellular spores was characteristic, while in the Tabernas (whether crusts or bare surface) these fungi mostly comprised less than one-third of the contribution index. On the other hand, the thermotolerant group of *Aspergillus* spp. (mostly *A. fumigatus*), was much more abundantly represented in the Tabernas crusts and bare surface as compared to the Negev and Tengger crusts.

In contrast to melanin-containing fungi, the distribution of aspergilli displayed a clear spatial pattern with significantly higher contribution in the more xeric habitats at the plain sun-exposed position, and at each position—in more edaphically harsh bare surface. Remarkably, at the north-oriented position in the cyanobacterial crusts and crusts dominated by *D. diacapsis*, the contribution of melanin-containing species was the highest, with *Aspergillus* spp. exhibiting simultaneously the lowest contribution. These microfungal groups also displayed the opposite relationship with electrical conductivity which in a definite interval may serve as an indirect indicator of various edaphic conditions being closely

associated with the amount of dissolved salts, organic matter content, and water holding capacity, soil texture, and porosity, e.g., Reference [49]. We therefore assume that xeric (melanized microfungi) and thermotolerant (aspergilli) groups may alternate in abundance in accordance with edaphic conditions. Similarly, in the extremely hot south-oriented sun-exposed habitats at the southern and central Negev Desert, not melanized species with large many-celled conidia, but *A. fumigatus* dominated microfungal communities [50,51]. As for the group of mesophilic *Penicillium* spp., it was predictably an eligible component of the communities and did not exhibit any consistent pattern in its spatial distribution.

Variations in density of microfungal isolates, which can be considered an indirect indication of fungal biomass, followed the expected distributional pattern. Significantly higher numbers of CFU characterized communities at the north-oriented position as compared to the more xeric communities at the plain sun-exposed position. Remarkably, isolate density displayed significant positive linear relationship with the chlorophyll content, similarly to the relationship found for the Negev crusts [19]. Due to the close link between chlorophyll content and organic matter [20], habitats with low organic matter content, such as bare surfaces, expectedly harbored the lowest numbers of CFU. Among crust types, the relatively low-biomass cyanobacterial crusts contained a significantly lower number of microfungal isolates than the lichen-dominated crusts. Isolate density displayed also significant and negative correlation with pH, which is explainable following the negative relationship established between organic matter content and pH values [52]. Additionally, at the plain sun-exposed position, the crusts dominated by the more xeric brown *P. decipiens* had substantially lower isolate density than the crusts dominated by the more mesic *S. lentighera* and *D. diacapsis*. On a whole, the distributional pattern of isolate densities found in the Tabernas crusts was similar to that of the Negev and Tengger deserts [19,22].

Spatial variations of diversity characteristics followed some consistent tendency, with the diversity level being significantly higher at the plain sun-exposed position as compared to the north-oriented positon. Moreover, communities in the cyanobacterial crusts at both positions were more heterogeneous and even than the communities from the lichen-dominated crusts. A similar pattern was found at NRS in the Negev Desert where the high-biomass moss-dominated crusts, which contained the highest density of microfungal isolates, had a lower number of species and diversity level of the communities than the cyanobacterial crusts [19]. The negative relationship between diversity level and CFU was rather expected. Often, an increase in the isolate density is caused by an abundant development of one or two species, which decrease both species richness and the diversity level of the microfungal communities.

The comparison of diversity level, composition, and structure of microfungal communities inhabiting the habitats at different positions showed that the communities at the plain sun-exposed position were much less variable than the communities at the north-oriented position. It is likely that more homogeneous and severe microclimatic conditions at the plain sun-exposed positon decrease the variability of microfungal communities, while more temperate and diverse abiotic conditions at the north-oriented positon support the development of more diverse and variable communities.

The type of surface—different crust types or bare surface—significantly affected the characteristics of microfungal communities. Similarly, in the Gurbantunggut Desert of Northwestern China, fungal communities were subjected to significant successional changes with biocrust development from bare sand, algal crusts, lichen crusts, to moss crusts [17]. Moreover, the surface type was found to influence the microfungal communities more strongly than the habitat position—plain sun-exposed or north-oriented. Following the close association between crust type and water availability [8,20], we may conclude that water availability not only determines the composition and biomass of the autotrophic components of the crusts but, expectedly, also influences the structure and biomass of the crust microfungal communities. Additionally, the fact that the interaction between the type of surface and the position of habitats in the Tabernas Desert significantly influenced the structure of microfungal communities and their species richness, emphasizes the ability of the communities to sensitively react to microscale variability of the environment.

## 5. Conclusions

The present study showed that the mycobiota of different crust types, as well as of bare surface, in the Tabernas Desert was dominated by melanin-containing species, similar to the majority of other desert mycobiota. However, unlike the studied crusted areas of the Negev and the Tengger deserts, melanized fungi with large multicellular spores were much less abundantly represented in the Tabernas crusts, while a thermotolerant group of fungi, *Aspergillus* spp., had a remarkable contribution to the community structure. The variations in crust composition, biomass, and the position of habitats were accompanied by the variations in microfungal community structure, diversity level, and isolate densities. The study revealed the important ecological regularities in structuring microfungal communities in crusted and noncrusted soil in the area. It showed that microclimatic factors differentiated by different position of the habitats and edaphic properties play an essential role in the development of these communities, and their structure can be a sensitive indicator of changing environmental conditions at a microscale.

The pattern revealed in this study referred only to the culturable fraction of fungal communities in the studied desert soils. However, so far, this fraction remains a significant and essential part of soil mycobiota both taxonomically and functionally [35]. Moreover, a comparison of the results obtained by different methodologies in arid regions showed that similar species belonging to the order Pleosporales prevailed both in the culturable mycobiota of biological soil crusts [19,22] and in the BSC mycobiota studied by the molecular technique [11–15]. Unfortunately, up to now, many fungi revealed by culture-independent approaches are known only from environmental DNA sequences creating so called "unnamed diversity" [53]. At the same time, culture-based methodology still allows important species-specific approach to fungal ecology with "leaning more biological details about fewer organisms and places" [54].

**Author Contributions:** Conceptualization, I.G., R.L. and G.I.K.; Resources, R.L.; Methodology, I.G. and G.I.K.; Investigation, I.G. and G.I.K.; Formal analysis, I.G.; Writing—original draft preparation, I.G.; Writing—review and editing, I.G., R.L. and G.I.K.

**Funding:** This research received no external funding.

**Acknowledgments:** We thank the Israeli Ministry of Absorption for the financial support of this research.

**Conflicts of Interest:** The authors declare no conflicts of interest.

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
