# Peer review of "Cultured Microfungal Communities in Biological Soil Crusts and Bare Soils at the Tabernas Desert, Spain"

_soilsystems, doi:10.3390/soilsystems3020036_

Round 1

Reviewer 1 Report

Review:  Cultured microfungal communities in biological soil 2 crusts and bare soils at the Tabernas Desert, Spain.

In this paper authors present a nice study about microfungal communities in a great system-a desert-and the influence of environmental factors (orientation/sun exposition and Surface type/crust presence) on fungal diversity and community structure. The study is interesting and of scientific interest. There are however a number of weaknesses, some of which could be addressed.

One flaw of the study is that the methodology used only provides a limited knowledge of the microfungal communities. Up-to date molecular technologies would surely improve the relevance of the results. Nevertheless, since authors admitted some limitations of the approach, I think the obtained results are interesting for the scientific community.

My main concern is how morphological identification of the colonies provides quantitative data. It is hard to know how this was performed because it is poorly explained in the methods section. What bibliography was used for taxonomy? Which fungal features were taken into account? What about microscopy? And, the most important, how quantitative data –abundance, number of isolates, data for Shanon index and so on- were obtained from colonies? I mean, was each single fungal colony identified? I find it hard to do in an accurate way, since 77 species were identified from a very large number of colonies as showed in figure 6.

Experimental design must be better described. Section 2.2. would improve with a broader description of experimental design and afterwards description of sampling.  If I get the idea, authors try to understand the effect of two factors on fungal communities. One is sun exposition/position/orientation (with two levels: north,south) and the other is the surface type/crust presence (with five levels: cianocrust, lichen S. lentigera crust, lichen D. diacapsis crust, lichen P. decipiens crust and bare soil).  

Results in subsections 3.2, 3.3, 3.4 would benefit from reorganization and reduction. It seems easier to follow by starting with density of microfungal isolates in figure 6, following with composition of fungal communities . i.e. the list of species and their abundance in table 2, then diversity parameters in table 1, then contribution of major fungal groups or relative abundance of major groups in figure 4, and finally the analysis of the tested factors (exposition/orientation/position and crust presence/surface ) on fungal communities together with PERMANOVA test described in lines 228-239.

I recommend authors try to use the same terminology throughout the paper, tables and figures; for instance north position, south position instead north-oriented, north-oriented localities, north-oriented slope, plain sun-exposed, which is confused. Or surface type instead crust type, crust presence, crust dominated by D. diacapsis, etc. Also use the same acronyms in tables or figures when necessary. (For example: Fig 3, Fig. 4, table 2 have different notations)

The figure legends are mixed and do not correspond with the data showed. Please correct the mistakes.

Other minor comments:

Lines 137-139 are unclear.  Nothing is said before or after about crusts by L.isidiata.

l. 187-188. Providing “contribution of each group to mycobiota” is equal to “relative abundance”  it would be clearer for readers to use relative abundance both in the text and corresponding tables.

l. 212. Please state Psora decipiens is the most xeric lichen crusts in material and methods instead here.

L 240. This sentence is confusing. Does that means 55 isolates from sun-exposed and 56 from north-oriented? Or 55 from crusts and 56 from bare soils?

l. 299-300. Please check, it sounds repetitive.

Why chlorophyll content is expressed mg/m2? There is not statistic variability of data in Figure 3. A figure legend showing the meaning of acronyms in X-axis would be welcome.

Table 1. Use R instead S for richness. As said before, please use the same notation for the Surface type and orientation throughout the manuscript, tables and figures.

It is not clear how contribution index (%) and relative abundances were calculated. As explained in lines 186-187 it was estimated as relative abundance in total isolate number in the sample (number of isolates of a particular group /total number of isolates).  If I have well understood, the value for each surface type should be rather lower than 100 providing other species were isolated.   However, in figure 4 the values almost reach 100 %. Please preferably use relative abundance instead contribution.   In the same sense, please check for data in table 2, the sum in each column cannot be higher than 100.

l. 408-410. The effect of microclimatic factor on microfungal communities has not been proven in this study. Please rewrite or delete.

Please revise language style, there are many unnecessary words or sentences. Just a few  examples:  l. 117 sizable diversity, l. 123 abounding in the area,  l. 336 all mycologically studied, l. 401 thus, l. 402 the majority of.

Rewrite abstract in accordance with the changes addressed.

Author Response

Reviewer 1

Review:  Cultured microfungal communities in biological soil 2 crusts and bare soils at the Tabernas Desert, Spain.

In this paper authors present a nice study about microfungal communities in a great system-a desert-and the influence of environmental factors (orientation/sun exposition and Surface type/crust presence) on fungal diversity and community structure. The study is interesting and of scientific interest. There are however a number of weaknesses, some of which could be addressed.

One flaw of the study is that the methodology used only provides a limited knowledge of the microfungal communities. Up-to date molecular technologies would surely improve the relevance of the results. Nevertheless, since authors admitted some limitations of the approach, I think the obtained results are interesting for the scientific community.

My main concern is how morphological identification of the colonies provides quantitative data. It is hard to know how this was performed because it is poorly explained in the methods section. What bibliography was used for taxonomy? Which fungal features were taken into account? What about microscopy? And, the most important, how quantitative data –abundance, number of isolates, data for Shanon index and so on- were obtained from colonies? I mean, was each single fungal colony identified? I find it hard to do in an accurate way, since 77 species were identified from a very large number of colonies as showed in figure 6.

-          All emerging colonies after examination under binocular (magnification of x45) were divided into morphotypes according to the colony appearance (texture, color, size, etc.) and whenever it was possible – presence of a definite kind of sporulation (fruit bodies or spores). Each morphotype was transferred to cultivation media for its purification and then the pure cultures were subjected to species identification based mainly on the microscopic examination of morphological characteristics of sporulation - fruit bodies and/or sporophores and spores: their shape, size, color, and mode of aggregation. The corresponding explanation has been added to the subsection 2.3. In the identification process, I followed the identification keys. On a whole, I have almost 40-yeared experience in the isolation and morphological identification of soil microfungi, and I have almost all possible classical and recent identification books including the revisions published in "Studies in Mycology", "Fungal Diversity", etc. for various fungal groups – it is hard to list all of them.

-          Relative abundance of species and Shannon index were calculated after the identification of species had been completed.

Experimental design must be better described. Section 2.2. would improve with a broader description of experimental design and afterwards description of sampling.  If I get the idea, authors try to understand the effect of two factors on fungal communities. One is sun exposition/position/orientation (with two levels: north, south) and the other is the surface type/crust presence (with five levels: cyanocrust, lichen S. lentigera crust, lichen D. diacapsis crust, lichen P. decipiens crust and bare soil). 

-          described 

Results in subsections 3.2, 3.3, 3.4 would benefit from reorganization and reduction. It seems easier to follow by starting with density of microfungal isolates in figure 6, following with composition of fungal communities. i.e. the list of species and their abundance in table 2, then diversity parameters in table 1, then contribution of major fungal groups or relative abundance of major groups in figure 4, and finally the analysis of the tested factors (exposition/orientation/position and crust presence/surface ) on fungal communities together with PERMANOVA test described in lines 228-239.

-          The subsections have been reorganized according to the reviewer suggestion. But we left the PERMANOVA test as a part of the subsection devoted to the composition of microfungal communities because this test compared namely their composition.

I recommend authors try to use the same terminology throughout the paper, tables and figures; for instance north position, south position instead north-oriented, north-oriented localities, north-oriented slope, plain sun-exposed, which is confused. Or surface type instead crust type, crust presence, crust dominated by D. diacapsis, etc. Also use the same acronyms in tables or figures when necessary. (For example: Fig 3, Fig. 4, table 2 have different notations)

-          In the corrected version of the manuscript, we used the same terminology throughout the text, tables and figures, namely, plain sun-exposed position and north-oriented position, as well as surface type - bare surface, cyanobacterial crusts, crusts dominated by (name of lichen species). We used the same acronyms in Figs. 3, 5, and Table 1, where the abbreviations of surface type had been used, and the same names of surface types in Table 2 and Figs. 4, 6. 

The figure legends are mixed and do not correspond with the data showed. Please correct the mistakes.

-          corrected

Other minor comments:

Lines 137-139 are unclear.  Nothing is said before or after about crusts by L.isidiata.

        -    The sentence has been deleted.

l. 187-188. Providing “contribution of each group to mycobiota” is equal to “relative abundance”  it would be clearer for readers to use relative abundance both in the text and corresponding tables.

       -  In order not to confuse between "contribution" and "relative abundance" according to the definition of those different terms in the subsection 2.5, we used them separately and consistently throughout the text.

l. 212. Please state Psora decipiens is the most xeric lichen crusts in material and methods instead here.

        -     corrected

L 240. This sentence is confusing. Does that means 55 isolates from sun-exposed and 56 from north-oriented? Or 55 from crusts and 56 from bare soils?

        -   It means that 55 species were isolated from the habitats at the plain sun-exposed position and 56 species were isolated from the habitats at the north-oriented position. The corresponding corrections have been made in the paragraph. 

l. 299-300. Please check, it sounds repetitive.

         -   corrected

Why chlorophyll content is expressed mg/m2? There is not statistic variability of data in Figure 3. A figure legend showing the meaning of acronyms in X-axis would be welcome.

-          The presentation of chlorophyll in mg/m2 is common. It can be found in many papers (see for instance: Brostoff et al., 2002 (Flora); Bu C et al-2014 (PLoS ONE); Büdel et al-2009 (Microb Ecol); Hawkes & Flechtner-2002 (Microbial Ecol); Jia et al-2014 (J Hydrol); Lan et al-2011 (SBB); Lange et al-1994b (Func Ecol)). Its advantage lies in the fact that photoautotrophic organisms require light and reside at the interface between the soil and the atmosphere. Therefore, their density and biomass is dependent on this shallow film of soil which enables the photoautotrophs light for photosynthesis. Moreover, as weight is a function of the bulk density of the soil, and different soils have different bulk densities, a comparison based on surface area will provide us a better mean for a comparison of various habitats/soils in different parts of the world.

-          Standard deviation has been added to the data in Fig. 3.

Table 1. Use R instead S for richness. As said before, please use the same notation for the Surface type and orientation throughout the manuscript, tables and figures.

-          corrected

It is not clear how contribution index (%) and relative abundances were calculated. As explained in lines 186-187 it was estimated as relative abundance in total isolate number in the sample (number of isolates of a particular group /total number of isolates).  If I have well understood, the value for each surface type should be rather lower than 100 providing other species were isolated.   However, in figure 4 the values almost reach 100 %. Please preferably use relative abundance instead contribution.   In the same sense, please check for data in table 2, the sum in each column cannot be higher than 100.

          -   We have added the explanation of how relative abundance of species was calculated to the first paragraph of the section 2.5. The second paragraph contains the explanation for the contribution index calculated as an average of relative abundance of a particular microfungal grouping in the sample and its percentage in the Shannon index, and why this index is preferable in comparison with the direct relative abundance data. Three microfungal groupings chosen for the study cover the great majority of the isolates, additionally, Aspergillus niger belongs both to melanin-containing species and Aspergillus spp.; because of that in Fig. 4 the sum of contribution indexes almost reaches 100%. The data in Table 2 (now Table 1) have been checked, the sum in each column is 100.

l. 408-410. The effect of microclimatic factor on microfungal communities has not been proven in this study. Please rewrite or delete.

        -  Different positions of the habitats (north-oriented and plain sun-exposed) in the studied area reflect the differences in microclimatic factors – temperature, insolation, and as a consequence – in water regime. Such effect is characteristic for the Negev Desert (Israel), where habitat location determines the type of crust and its biomass (Kidron et al., 2009, 2010; Kidron and Vonshak, 2012). Because of that we assumed that microclimatic factors played an essential role in the development of studied microfungal communities. The corresponding corrections have been made in Conclusion

Please revise language style, there are many unnecessary words or sentences. Just a few example:  l. 117 sizable diversity, l. 123 abounding in the area,  l. 336 all mycologically studied, l. 401 thus, l. 402 the majority of.

-          Language of the paper has been checked and corrected.

Rewrite abstract in accordance with the changes addressed.

-          The abstract has been corrected accordingly.

All corrections are highlighted in yellow in the text.

Reviewer 2 Report

The authors described microfungal communities in biological soil crusts in comparison to bare soils. They examined the microfungal structure, density and diversity is above mentioned soils and also they preformed basic soil properties characterization.The results showed that Aspergillus spp. was dominant in microfungal community structure. The methods and analyses are well described, appropriate, and the results are clearly presented, however it would be interesting to perform also molecular analyses of microfungal community by next generation sequencing or also to provide the results of specific properties of fungal strains from collected soil samples.

The topic of this manuscript is focused on not wide groups of researchers, especially with only methods proposed by Authors. The practical or other context of the research should be explained or underlined, because this study are dedicated to small group of scientists.

The conclusions should be formulated based on the research, and this part should not only be a discussion with other results and study. You should underline innovative aspect of the study not only write that the results confirm the same results that was obtained in previous research of the other authors.

I recommend publication of the manuscript after minor revision.

Author Response

Reviewer 2

The authors described microfungal communities in biological soil crusts in comparison to bare soils. They examined the microfungal structure, density and diversity is above mentioned soils and also they performed basic soil properties characterization. The results showed that Aspergillus spp. was dominant in microfungal community structure. The methods and analyses are well described, appropriate, and the results are clearly presented, however it would be interesting to perform also molecular analyses of microfungal community by next generation sequencing or also to provide the results of specific properties of fungal strains from collected soil samples.

The topic of this manuscript is focused on not wide groups of researchers, especially with only methods proposed by Authors. The practical or other context of the research should be explained or underlined, because this study are dedicated to small group of scientists.

The conclusions should be formulated based on the research, and this part should not only be a discussion with other results and study. You should underline innovative aspect of the study not only write that the results confirm the same results that was obtained in previous research of the other authors.

-          The innovative ecological aspect of the study has been underlined in Conclusion.

I recommend publication of the manuscript after minor revision.

Round 2

Reviewer 1 Report

The paper has been considerably improved as the changes suggested have been addressed.

Just one comment remains:   one or two references used for taxonomic identification of the colonies should be included.

Author Response

Just one comment remains:   one or two references used for taxonomic identification of the colonies should be included.

   - Three references used for taxonomic identification have been included in the subsection 2.3.